

# Decoupling social status and status certainty effects on health in macaques: a network approach

Jessica J. Vandeleest[1,2,*], Brianne A. Beisner[1,2,*], Darcy L. Hannibal[1,2], Amy C. Nathman[2], John P. Capitanio[2], Fushing Hsieh[3], Edward R. Atwill[1] and Brenda McCowan[1,2]

[1] Population Health & Reproduction, University of California, Davis, California, United States
[2] Brain, Mind and Behavior, California National Primate Research Center, Davis, California, United States
[3] Department of Statistics, University of California, Davis, California, United States
* These authors contributed equally to this work.

## ABSTRACT

**Background:** Although a wealth of literature points to the importance of social factors on health, a detailed understanding of the complex interplay between social and biological systems is lacking. Social status is one aspect of social life that is made up of multiple structural (humans: income, education; animals: mating system, dominance rank) and relational components (perceived social status, dominance interactions). In a nonhuman primate model we use novel network techniques to decouple two components of social status, dominance rank (a commonly used measure of social status in animal models) and dominance certainty (the relative certainty vs. ambiguity of an individual's status), allowing for a more complex examination of how social status impacts health.

**Methods:** Behavioral observations were conducted on three outdoor captive groups of rhesus macaques (N = 252 subjects). Subjects' general physical health (diarrhea) was assessed twice weekly, and blood was drawn once to assess biomarkers of inflammation (interleukin-6 (IL-6), tumor necrosis factor-alpha (TNF-α), and C-reactive protein (CRP)).

**Results:** Dominance rank alone did not fully account for the complex way that social status exerted its effect on health. Instead, dominance certainty modified the impact of rank on biomarkers of inflammation. Specifically, high-ranked animals with more ambiguous status relationships had higher levels of inflammation than low-ranked animals, whereas little effect of rank was seen for animals with more certain status relationships. The impact of status on physical health was more straightforward: individuals with more ambiguous status relationships had more frequent diarrhea; there was marginal evidence that high-ranked animals had less frequent diarrhea.

**Discussion:** Social status has a complex and multi-faceted impact on individual health. Our work suggests an important role of uncertainty in one's social status in status-health research. This work also suggests that in order to fully explore the mechanisms for how social life influences health, more complex metrics of social systems and their dynamics are needed.

Corresponding author
Jessica J. Vandeleest,
vandelee@ucdavis.edu

## INTRODUCTION

Social life influences mental and physical health (*Thoits, 2011*; *Nunn et al., 2015*). For example, a lack of satisfactory social relationships has been shown to be associated with poor health and high quality relationships can buffer individuals from stress (*Hostinar, Sullivan & Gunnar, 2014*; *Hawkley & Capitanio, 2015*). Treatment and prevention of illness thus requires a detailed understanding of the complex interplay between social and biological systems. Although past research has clearly shown that the social environment plays a critical role in shaping health, the effect of the complex and multi-scale dynamic nature of social relationships on health remains poorly understood (*Thoits, 2011*). For example, although the absence of social relationships has been associated with poor health outcomes, social life consists of more than just the presence or absence of social relationships. Qualities of these relationships, such as stability and role within the community, are also important factors (*Sapolsky, 1992*; *Thoits, 2011*). Therefore, approaches that empirically recognize the inherent complexity of social life are critical to improve our understanding of how social life influences health *McCowan et al. (2016)*. In this paper, we use a nonhuman primate model to investigate an understudied aspect of social status and its effects on health outcomes. We use a measure of status certainty to test whether greater uncertainty in status relationships is coincident with greater levels of inflammation and poor health outcomes.

Status is one component of social life whose impact on health has been widely studied. While in humans there is a clear general pattern that individuals of low status (i.e. socioeconomic status) often have greater disease incidence and shorter lifespans (*Adler et al., 1994*; *Marmot & Sapolsky, 2014*; *Chetty et al., 2016*), across species the impact of status on health is less clear (*Creel, 2001*; *Habig & Archie, 2015*). For example, in species such as baboons, macaques, marmots, and meerkats, low status individuals are at greater risk of poor health outcomes, such as poor cardiovascular health, reduced immune function, and higher levels of glucocorticoids (GCs) (*Sapolsky & Mott, 1987*; *Sapolsky & Share, 1994*; *Shively & Clarkson, 1994*; *Hackländer, Möstl & Arnold, 2003*; *Young et al., 2006*; *Archie, Altmann & Alberts, 2012*). While GCs are not a health outcome, they are frequently used as a biomarker for increased risk for negative health outcomes because they are released in response to social stress and play an important role in regulating immune function (*Sapolsky, Romero & Munck, 2000*). In contrast, the potential costs of high status center on GCs and parasite loads. In social carnivores and many cooperative breeding species, high status individuals tend to have higher GCs than low status individuals (summarized in: *Creel (2001)* and *Creel et al. (2013)*). In addition, high status individuals across a variety of species have been shown to experience higher parasite loads than low status individuals (*Habig & Archie, 2015*). Compounding this confusion are species in which status effects differ by sex or study population (*Schoech, Mumme & Moore, 1991*; *Creel, MarushaCreel & Monfort, 1996*; *Arnold & Dittami, 1997*) as well as

species in which no status effects on health have been found (*Mays, Vleck & Dawson, 1991*; *Wingfield, Hegner & Lewis, 1991*; *Lynch, Ziegler & Strier, 2002*). Recently, researchers have attempted to reconcile these differences to examine broad associations between social status and health across species. For example, a recent meta-analysis examining the impact of status on immune parameters and parasite load found little evidence for consistent effects of status on immune function but did find that high status males are at greater risk for parasitic infections (*Habig & Archie, 2015*).

The lack of cross species consensus on the impact of social status on health may be partly due to the fact that social status is more complex than a simple linear ranking of individuals based upon income, education level, or, in the case of nonhumans, dominance. The advancement of novel network techniques provides an opportunity to begin measuring the complexity of social life in new ways *McCowan et al. (2016)*. Social networks have been shown to impact both mental and physical health outcomes, highlighting the utility of these methods in understanding population health (*Pachucki et al., 2015*; *Perkins, Subramanian & Christakis, 2015*). One understudied, and potentially critical, aspect of status is the relative stability and predictability associated with one's status. In the human literature, unpredictability in access to resources (e.g. food, medical care, and housing) and job insecurity have been suggested to be features of low socioeconomic status that contribute to poorer health outcomes (*Adler et al., 1994*). In the animal literature, group-level instability of the hierarchy and dominance rank reversals have similarly been suggested to influence patterns of association between social status and health (e.g. *Sapolsky, 1992*; *Muller & Wrangham, 2004*; *Sapolsky, 2005*). Even within a stable social group, individual-level social relationships are dynamic and can vary in stability (*Hinde, 1976*). For example, while a change in rank may be stressful for both individuals involved, the negative impact on health is often greater for the animal that loses rank compared to the animal that gains rank (e.g. *Sapolsky, 1992*; *Shively & Clarkson, 1994*). This body of work highlights that the impact of status on health may depend not only on one's position in the hierarchy, but also on the stability and predictability of one's status relationships. Therefore, we propose that a measure of social status that quantifies such individual-level instability or uncertainty may explain cross-species differences regarding the impact of status on health outcomes.

We use a nonhuman primate model to disentangle two components of status, dominance rank (linearly ordered status relative to other animals in the group) and dominance certainty (the probability that status is decided and stable), using a novel computational network-based approach called Percolation and Conductance (*Fushing, McAssey & McCowan, 2011*; K. Fujii et al., 2014, unpublished data). This method characterizes the flow of status (i.e. the overall direction of aggression and submission) through pathways in the network and each individual's fit within that hierarchical flow to quantify both dominance rank and dominance certainty. Like most methods for measuring dominance rank, Percolation and Conductance uses direct aggression and/or submission data to create a dominance hierarchy. Unlike other methods, however, it uses information from *indirect pathways* through the network of aggressive interactions to modify the likely rank association between individuals and to measure the consistency of information flow

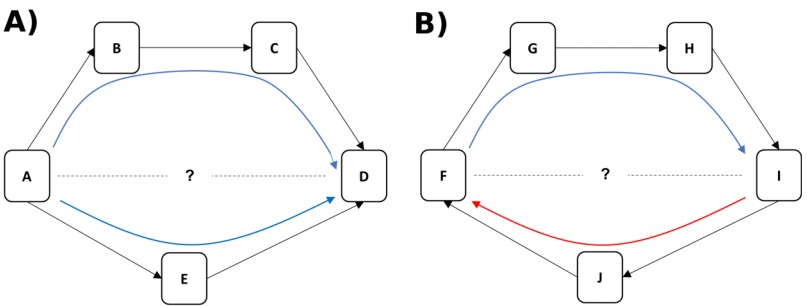

**Figure 1 Inference of dominance rank and certainty using a network.** (A) Although animals A and D do not directly interact, it can be inferred that A outranks D through the indirect pathways in the network. Certainty for this inference is increased when multiple pathways flow in the same direction (i.e. from A to D). (B) Although animals F and I do not interact, it can be inferred through the most direct pathway (through individual J) that I outranks F. Certainty for this inference, however, is lower due to the contradictory flow of dominance from F to I (through individuals G and H).

through the network (*Fushing, McAssey & McCowan, 2011*). For example, if animal A directs aggression at animal E (and E submits) and E directs aggression at D (and D submits), we can infer that A likely outranks D even if they have never been observed to interact (see Fig. 1A). Greater consistency in the direction of dominance pathways from A to D results in higher certainty that A outranks D (Fig. 1A), whereas evidence of inconsistent direction (e.g. between F and I in Fig. 1B) reflects dominance ambiguity. Our method thus solves the problem of sparse or missing data in the win/loss matrix (e.g. the treatment of zeroes in the matrix is a non-trivial issue: (*de Vries, 1995*)) by using these dominance pathways as additional sources of information about each pairwise dominance relationship. Our method additionally provides a measure of how well the direction of each individual's dominance interactions fit, on average, within the larger group-level pattern of aggression from dominants to subordinates, a measure we call dominance certainty.

Our study aimed to examine the relative impacts of dominance rank and dominance certainty on biomarkers of inflammation and diarrhea, with particular interest in whether individual level dominance certainty either better explains variation in health or moderates the rank-health association. Similar to results reported in baboons and long-tailed macaques (*Sapolsky & Share, 1994*; *Shively & Clarkson, 1994*), we expected to find that low-ranked individuals exhibited poorer health than high-ranked individuals. We predicted, however, that reduced certainty in dominance relationships may be associated with poorer health outcomes either independent of rank or specifically among individuals that stand to lose rank, inasmuch as uncertainty in one's relationships is likely to be stressful. We chose to include both biomarkers of immune function as well as a general health outcome because previous research suggests that effects of status are not always the same across health measures (*Habig & Archie, 2015*).

## MATERIALS AND METHODS

### Subjects and housing

The subjects of this study were 252 rhesus macaques (71 males, 181 females; age range: 3–29 years, mean = 7.7 years) from three outdoor social groups (126–185 individuals

per group) that were part of the breeding colony at the California National Primate Research Center (CNPRC) in Davis, CA. Animals were socially housed in 0.2 ha outdoor enclosures containing multiple A-frame structures, suspended barrels, swings, and perches and were free to interact as they chose. Animals were fed a standard diet of monkey chow twice per day at approximately 0700 h and between 1430 and 1530 h. Fresh fruit or vegetables were provided one time per week and seed mixture provided daily. Water was available ad libitum. Animals housed in outdoor enclosures were managed with a minimum level of disturbance. At the end of the study all subjects were relinquished to the CNPRC breeding colony. This research was approved by the University of California Davis Institutional Animal Care and Use Committee.

## Rank and dominance certainty data collection

Each group was observed for 5–7 weeks, one group in spring (Group A: March–April 2013) and two in fall (Group B: September–October 2013, Group C: September–October 2014) as part of a larger study on social networks and health. Unexpected management events resulted in lengthening (e.g. Group A) or shortening (e.g. Group B) the six-week study period by a few days in certain groups. We used an event sampling design to collect all instances of aggressive and submissive interactions. Two observers collected data for 6 h on 4 days per week from 0900–1200 and 1300–1600 h. Aggressive and submissive events were recorded as a series of dyadic interactions. A total of 13,567 events were recorded during 444 h of observation. Aggression was categorized according to severity and included threat (open mouth stare, brow flash, ear flap), mild aggression (threat and follow, lunge, push, slap, chase < 6 meters), moderate aggression (grapple, wrestle, chase > 6 meters), and intense aggression (pin or bite). Submission categories included freeze/turn away, move away, run away < 6 meters, run away > 6 meters, and crouch. Data on all dyadic aggressive interactions were used for constructing aggression networks to calculate dominance rank and certainty.

Dominance rank and certainty were calculated using the *Perc* package in R (*Fujii et al., 2015*) which uses a new network-based approach that combines information from direct dominance interactions with information from multiple indirect dominance pathways (via common third parties) to quantify dyadic dominance relationships (*Fushing et al., 2011*; *Fujii et al., 2015*). Essentially, individuals with dominance pathways that run counter to the primary direction of aggression (or submission) in the hierarchy have more ambiguous status than individuals who do not. The method begins by using a modified random walk algorithm to exhaustively identify all directed pathways in the network (e.g., A → B ← C is not a directed pathway, but A → B → C is). To determine how much to weight the imputed (i.e. indirect) 'wins' from these pathways, the transitivity of the network is calculated as the proportion of transitive triangles (as opposed to cyclic or nontransitive triangles). In networks with high transitivity such as ours (> 95% of triangles are transitive, such that A > B > C and A > C), a 'win' that is imputed from an indirect pathway is more likely to reflect the true direction of dominance, and thus given higher weight, than in a network with lower transitivity. Regardless, wins from direct interactions are always weighted more than imputed 'wins'

from network pathways. Finally, given the number of direct wins and imputed 'wins' from pathways, we calculate the probability that the row animal is dominant to the column animal using a Beta distribution to incorporate a source of statistical probability that reflects the level of uncertainty expected given the current data on each pair (for a more detailed explanation, see: *Fushing, McAssey & McCowan, 2011*). The matrix of these dyadic dominance probability values (range: 0–1) thus represent the cumulative information from all network pathways between each pair of animals. A dyadic dominance probability of one reflects the highest possible certainty that the row animal outranks the column animal, whereas 0.5 means the dominance relationship is perfectly ambiguous. The matrix of dyadic dominance probabilities was then used to generate the lowest cost linear rank order (see *Fushing et al., 2011* for details).

Sparse and missing data (i.e. pairs of animals that are either infrequently or never observed to interact) are a common problem in animal behavior, and most ranking methods (such as the IS&I method (*de Vries, 1998*), Elo-rating (*Neumann et al., 2011*; *Viswanath, 2016*) and the Bradley-Terry model: *Boyd & Silk (1983)* and *Albers & de Vries (2001)*) are vulnerable to sparse data. The Percolation and Conductance method addresses the problem of sparse and missing data by gathering dominance information from network pathways. For example, in our study group A only 42.3% of all possible dyads had at least one agonistic interaction in the network. The remaining dyads were never observed to interact, making estimation of their pairwise dominance relationships from direct interactions prone to error. Furthermore, of the 42.3% that did interact, they averaged less than two interactions per dyad (x = 1.89). Adding dominance *pathways* dramatically increased the information per dyad to an average of 207.4 dominance pathways across all pairs (using up to 3-step pathways: A→B→C→D). Given that transitivity is very high, the dominance information from pathways has a high probability of agreeing with the true relationship. Furthermore, it is likely that macaques have the cognitive capacity to use this dominance information because many social vertebrates are capable of transitive inference such as deducing A > C from A > B and B > C (*McGonigle & Chalmers, 1977*; *Davis, 1992*; *Bond, Kamil & Balda, 2003*; *Grosenick, Clement & Fernald, 2007*).

From these dyadic dominance probabilities (which contain both rank direction and certainty information), an average dominance probability was calculated for each subject to provide a metric of the overall certainty of each animal's rank. Prior to averaging, we transformed the dyadic dominance probabilities (initially bounded by 0–1) to range between 0.5 (ambiguous) and 1.0 (certain), thereby focusing on the information about certainty and ignoring rank direction. We also transformed ordinal dominance ranks for each group (derived from permuting the rows and columns of the dominance probability matrix) into the proportion of animals outranked within their respective groups (i.e. 0 is the lowest ranked animal and 1 is the highest ranked animal) to account for differences in group size. For the purposes of graphical representation, individual-level average dominance certainty was categorized into high, moderate, and low certainty categories for each group. We examined empirical distributions of average dominance certainty values (performed on each group separately) using the *segmented* package in

R to identify logical break-points for these categories (*Muggeo, 2008*). Groups differed in their break-points due to differences in distribution, and these group-specific categorizations of low vs. moderate vs. high dominance certainty are thus reflected in our plots.

## Social network measures and independence

Statistical analysis of social network metrics requires some attention to the lack of independence in network data. The calculation of a network metric for one node (e.g., an individual) typically involves some of the same edges as the same calculation for other nodes (*Croft, James & Krause, 2008*). As a result, a change in the value of one node could potentially affect the values of other nodes. We experimentally verified that this lack of independence was not an issue in this study by recalculating network measures after removing single individuals from the network. We selected three individuals with differing betweenness centrality (i.e. highest, median, and lowest) and experimentally removed each individual from the aggression network, one-at-a-time, recalculating the values of the remaining individuals each time, for a total of three recalculations (one for each individual). We chose to examine betweenness centrality because it is defined by shortest paths between all pairs in the network, similar to the counting of all paths between pairs in our Percolation and Conductance method. Bivariate and Spearman rank order correlations between the original dominance certainty and the new values indicate very little change (bivariate pairwise correlations for (a), (b) and (c): $0.999 \geq r \geq 0.932$; Spearman pairwise correlations: $0.998 \geq r \geq 0.941$). Furthermore, our Percolation and Conductance method incorporates statistical probability/uncertainty into the calculation of each dyadic dominance probability value, which serves to dampen any effect of dependence in the data.

## Blood sample collection

Blood samples were collected on a single day (between 0800–1200 h) during the fifth week of the 5–7 week observation period for each group using the CNPRC's standard method for biannual physical examinations. On sampling days, all animals in the group were immobilized (10 mg/kg of ketamine) and given standard physical examinations by veterinary staff (e.g., checked for injuries, weighed, assessed for pregnancy). Blood samples (5 mL) were also obtained from the femoral vein and serum was aliquoted and stored at −80 °C for later assay. Due to the large number of animals, blood was collected in batches of ∼15 samples. Batches were labeled with collection start and end times and animal identity to track the order of sample processing and control for effects of the capture and sedation procedure for animals processed later in the morning.

## General health assessments

Health indicators were recorded twice weekly by one observer between 0900–1200 and 1300–1600 h. The observer located and visually inspected each study subject in the group and scored presence/absence of liquid stool (i.e. observing defecation of liquid stool or observing fresh liquid fecal matter on the tail or rump). From these data, we counted the

total frequency of bouts of diarrhea per subject across the 5–7 week observation period. A bout of diarrhea was defined as either a single observation day with liquid stool or multiple consecutive observation days with liquid stool with the end of a bout marked by at least one observation day with no evidence of liquid stool. Occasionally animals were temporarily removed from the cage for veterinary care and were not available for health observations—these absences were recorded to control for total observation days.

## Pro-inflammatory proteins

We measured three pro-inflammatory proteins (interleukin-6 (IL-6), tumor necrosis factor-alpha (TNF-α), and C-reactive protein (CRP)) from frozen serum. These proteins were chosen because they are markers of general systemic inflammation and are demonstrated risk factors for multiple diseases (e.g. type 2 diabetes and atherosclerosis, *Pradhan et al., 2001*; *Libby, Ridker & Maseri, 2002*).

## Cytokine assay

Serum levels of IL-6 and TNF-α were measured using commercially available, species specific Milliplex multi-analyte profiling (MAP) reagents purchased from EMD/Millipore (Billerica, MA, USA), and utilizing Luminex Xmap technology (Luminex, Austin, TX, USA). Color coded polystyrene microbeads coated with specific antibodies for IL-6 and TNF-α were incubated with the serum samples, washed, and then were further reacted with biotinylated detector antibodies followed by Streptavidin-PE to label the immune complexes on the beads. After a final washing to remove all unbound material, the beads were interrogated in a BioPlex dual laser (BioRad, Hercules, CA, USA). The median fluorescent index for each sample was compared to a standard curve to calculate the concentration. Samples were tested in duplicate and had an intra-assay coefficient of variability of 15.3%. Samples falling below the threshold sensitivity of the assay (1.6 pg/mL) were assigned a value of one.

## C-reactive protein assay

Concentrations of CRP were determined using a latex particle immunoturbidmetric method on the Beckman Coulter AU480 clinical chemistry analyzer.

## Data analysis

Data analysis proceeded through a two-step process. First, hypothesized models (see below and Table 1) were tested for four dependent variables including three pro-inflammatory proteins (CRP, TNF-α, and IL-6) and one general health outcome (frequency of diarrhea bouts). The sets of hypotheses that guided model-fitting were explored for the following reasons. Age and sex were included as main effects in all models because both have been previously found to influence health (*Klein, 2000*; *Sansoni et al., 2008*). The impact of rank on health is also known to vary by sex in some populations (*Creel, MarushaCreel & Monfort, 1996*; *Abbott et al., 2003*), and it is reasonable to expect that the impact of rank may manifest differently at different ages. Therefore, sex by rank and age by rank interaction terms were also explored. We also examined a sex by dominance certainty interaction because the inherent structural

**Table 1  Hypotheses.**

| Hypothesis | Question | Variables |
|---|---|---|
| H0 | Null model | Y = control variables |
| H1 | Does health differ by age or sex class? | Y = sex + age |
| H2 | Does rank influence health beyond effects of age and sex? | Y = sex + age + rank |
| H3 | Does dominance certainty influence health beyond effects of age and sex? | Y = sex + age + DC |
| H4 | What are the relative impacts of rank and dominance certainty on health? | Y = sex + age + rank + DC |
| H5 | Does the impact of rank on health depend upon dominance certainty? | Y = sex + age + rank + DC + rank*DC |
| H6 | Does the impact of rank differ for juveniles, adults and geriatric animals? | Y = sex + age + rank + rank*age |
| H7a | Does the impact of status (i.e. rank) differ for males and females? | Y = sex + age + rank + rank*sex |
| H7b | Does the impact of status (i.e. dominance certainty) differ for males and females? | Y = sex + age + rank + DC*sex |
| H8 | Due to sex differences in how status is attained, does the interaction of rank and DC affect males and females differently? | Y = sex + age + rank + DC + sex*rank + sex*DC + rank*DC + sex*DC*rank |

**Notes:**
All models include a random effect of cage.
DC, dominance certainty.

differences in male vs. female rank acquisition (i.e. individual-level features such as age and body size drive male rank: (*Dittus, 1975*; *Sprague, 1992*; *Sprague, 1998*; *Sebastian, 2015*); family-level features such as agonistic support drive female rank: (*Sade, 1972*; *Datta, 1986*)) suggest that ambiguity may arise more readily amongst males than females and the presence of ambiguous relationships may have greater costs for females than males.

All data were analyzed using generalized linear mixed-effects regression models (*McCullagh & Nelder, 1989*; *Raudenbush & Bryk, 2002*) with group as a random effect. Models were run using a negative binomial distribution for IL-6, TNF-α and diarrhea bouts, and a gamma distribution for CRP. Appropriate distribution(s) for each outcome were chosen based on descriptive statistics, histograms, and Q-Q plots. We also evaluated the goodness of fit of these distributions using the Pearson chi-square statistic (*SAS Institute Inc., 2009*). We also included variables to control for known confounds: sample collection order for blood-based measures, and total health observation days for diarrhea. Due to the unavailability of blood samples for some animals, the cytokine and CRP analyses were run on 234 of the 252 subjects. Finally, ten of the study subjects showed CRP levels above 10 (a sign of possible infection), and these animals were included in analyses because they represent an important part of the health continuum. We note, however, that excluding these subjects from analyses did not change the magnitude or direction of the effects.

We used an Information Theoretic approach to determine which variables best explained each of our health outcomes. First, we ran all mixed-effects regression models to address our complete set of hypotheses outlined in Table 1. For each model, we present AICc scores, dAICc, model likelihoods (L = exp(−(1/2 ∗ dAICc))), Akaike weights, and evidence ratios (ratio = weight of best model/weight of comparison model) as outlined in *Burnham, Anderson & Huyvaert (2011)*. We then used model weights and dAICc to select a set of candidate models for each outcome variable. When the weight of the best

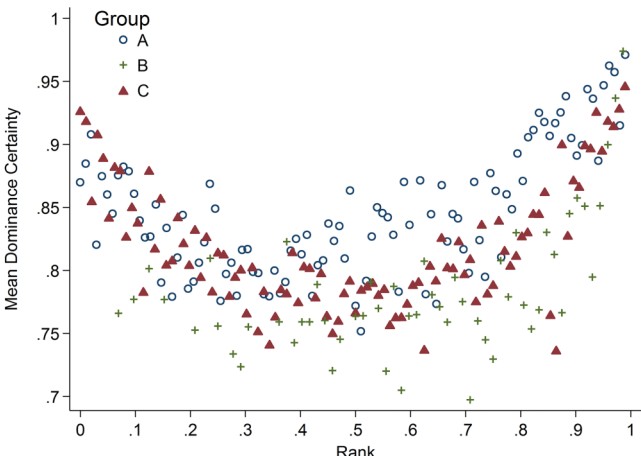

**Figure 2 Rank and dominance certainty.** Scatter plot of dominance rank and dominance certainty. Markers indicate group membership.

model was < 0.90 (*Burnham & Anderson, 2002*; *Symonds & Moussalli, 2011*), we included in the candidate set those models with dAICc ≤ 5 and discuss the inferences based on all models in this candidate set. Analyses were performed in SAS 9.4 and R 3.3.1 (*R Core Team, 2013*). Plots were produced in Stata 14.1. Although plots of model results are typically generated from marginal effects tables, this was not appropriate for our data due to the presence of discontinuous regions. We have used the alternative method of generating model specific plots from predicted values (*Hardin & Hilbe, 2007*; *Rabe-Hesketh & Skrondal, 2012*).

## RESULTS

### Rank and dominance certainty

Rank and dominance certainty were associated in a nonlinear manner (Fig. 2). Specifically, animals of high- and low-rank exhibited higher dominance certainty than those in the middle of the rank distribution, a pattern that held true for all three study groups. In addition, there was variability in dominance certainty at each level of rank. Specifically, dominance certainty ranged from 0.70–0.97 and 0.72–0.93 among the highest and lowest ranked tertiles of the sample, respectively, which is very similar to the range for the sample as a whole (0.70–0.97).

### Pro-inflammatory measures

Results for CRP indicated a single model with the highest weight (w = 0.93, H8; Table 2) in which the relationship between rank and CRP was dependent on dominance certainty and sex. High-ranking males with low dominance certainty had higher CRP, whereas little to no effect of rank was found in males with high dominance certainty (Table 3; Fig. 3A). For males with highly certain dominance relationships, increasing rank by 0.25 (moving up a quartile in rank) was associated with a *reduction* in CRP levels by 1.17 times. In contrast, for males with low dominance certainty, increasing in rank by 0.25 was associated with a 1.80 times *increase* in CRP. In females there was no change in

**Table 2 Model fitting for C-reactive protein.**

| | Model | AICc | ΔAICc | Model likelihood | Model weight | Evidence ratio |
|---|---|---|---|---|---|---|
| H8 | Y = SO + IL-6 + sex + age + rank + DC + rank*DC + rank*sex + DC*sex + sex*rank*DC | 1,022.74 | 0.00 | 1.000 | 0.930 | 1.00 |
| H5 | Y = SO + IL-6 + sex + age + rank + DC + rank*DC | 1,029.58 | 6.84 | 0.033 | 0.030 | 30.55 |
| H7a | Y = SO + IL-6 + sex + age + rank + DC + rank*sex | 1,030.86 | 8.12 | 0.017 | 0.016 | 57.94 |
| H4 | Y = SO + IL-6 + sex + age + rank + DC | 1,031.87 | 9.13 | 0.010 | 0.010 | 95.83 |
| H3 | Y = SO + IL-6 + sex + age + DC | 1,032.62 | 9.88 | 0.007 | 0.007 | 139.82 |
| H7b | Y = SO + IL-6 + sex + age + rank + DC + DC*sex | 1,033.29 | 10.55 | 0.005 | 0.005 | 195.28 |
| H1 | Y = SO + IL-6 + sex + age | 1,036.40 | 13.66 | 0.001 | 0.001 | 923.19 |
| H6 | Y = SO + IL-6 + sex + age + rank + DC + rank*age | 1,037.00 | 14.26 | 0.001 | 0.001 | 1,246.43 |
| H2 | Y = SO + IL-6 + sex + age + rank | 1,037.85 | 15.11 | 0.001 | 0.000 | 1,910.99 |
| H0 | Y = SO + IL-6 | 1,044.31 | 21.57 | 0.000 | 0.000 | 48,356.29 |

Notes:
Random effect: Cage; N = 234; SO, Sampling order; DC, Dominance certainty.

**Table 3 Model coefficients and SEs from the sets of candidate models for pro-inflammatory proteins.**

| | CRP (H8) coeff (SE) | IL-6 (H5) coeff (SE) | IL-6 (H8) coeff (SE) | IL-6 (H3) coeff (SE) | IL-6 (H1) coeff (SE) | TNF-α (H5) coeff (SE) |
|---|---|---|---|---|---|---|
| dAICc | 0 | 0 | 0.62 | 3.54 | 3.90 | 0 |
| Intercept | 2.48 (2.34) | −3.71 (3.15) | 0.65 (4.74) | 4.40 (1.23) | 2.46 (0.20) | −6.34 (3.87) |
| Rank | −0.08 (3.10) | 12.6 (4.48)* | 5.89 (6.23) | – | – | 22.0 (5.71)* |
| DC[1] | −1.79 (2.86) | 7.35 (3.81)+ | 1.98 (5.79) | −2.46 (1.53)+ | – | 13.9 (4.70)* |
| Sex[2] | −5.70 (3.12)+ | 0.034 (0.19) | −8.79 (6.58) | 0.18 (0.18) | 0.18 (0.18) | −0.4 (0.25) |
| Rank × DC | 0.11 (3.72) | −15.0 (5.30)* | −6.88 (7.44) | – | – | −26.2 (6.75)* |
| Rank × Sex[2] | 13.2 (4.57)* | – | 19.3 (9.51)* | – | – | – |
| DC × Sex[2] | 6.75 (3.77)+ | – | 10.7 (7.96) | – | – | – |
| Rank × DC × Sex[2] | −14.9 (5.41)* | – | −23.5 (11.2)* | – | – | – |
| IL-6 | 0.004 (0.002) | – | – | – | – | – |
| SO[3] | 0.026 (0.02)+ | 0.053 (0.03)+ | 0.054 (0.03)+ | 0.051 (0.03)+ | 0.041 (0.03) | 0.061 (0.03)+ |
| Age[4]-3 years | −0.004 (0.12) | −0.66 (0.25)* | −0.68 (0.25)* | −0.64 (0.23)* | 0.65 (0.23)* | −0.47 (0.34) |
| Age-4–5 years | −0.046 (0.11) | −0.88 (0.21)* | −0.88 (0.21)* | −0.82 (0.21)* | −0.82 (0.21)* | −0.83 (0.28)* |
| Age-13+ years | 0.27 (0.13)* | 0.24 (0.25) | 0.31 (0.25) | 0.16 (0.25) | 0.11 (0.26) | 0.33 (0.35) |

Notes:
[1] DC, Dominance certainty.
[2] Males were referent category.
[3] Sampling order.
[4] Adults (6–12 years) were the referent category.
* p < 0.05.
+ p < 0.1.

CRP with increasing rank at any level of dominance certainty (Table 3; Fig. 3B). Consistent with predictions, older animals also had higher levels of CRP than adults, subadults, or juveniles (Table 3).

For IL-6 two models with similar weight accounted for nearly 80% of total weight (Table 4), and both models provided evidence that the effect of rank on IL-6 levels was dependent on dominance certainty (w = 0.45, H5; Fig. 4) and also potentially on sex

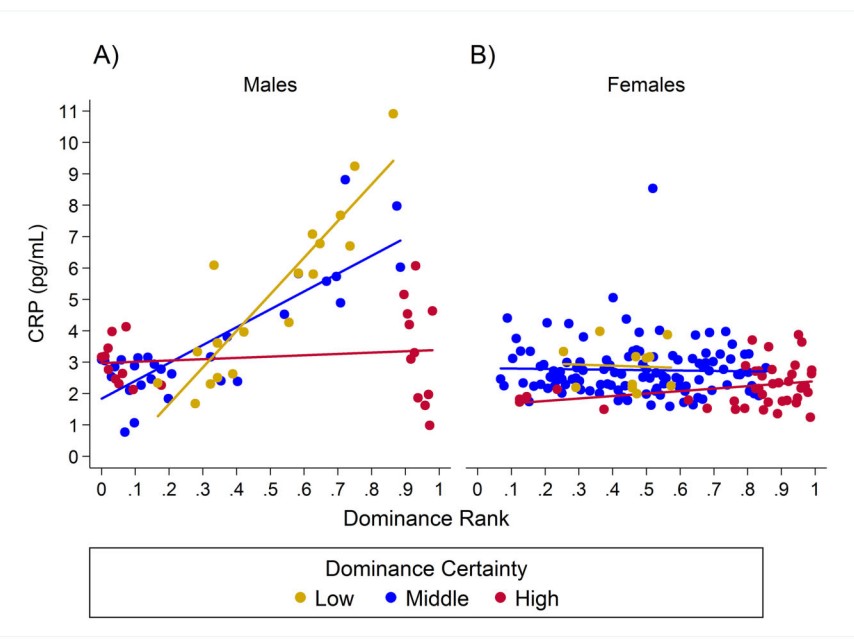

**Figure 3 Sex differences in the impact of dominance certainty and rank on CRP.** Predicted values of CRP. Panels A and B depict effects for CRP for males and females, respectively (based on model H8). Separate lines represent the interaction between dominance rank and dominance certainty.

**Table 4 Model fitting for IL-6.**

| | Model | AICc | ΔAICc | Model likelihood | Model weight | Evidence ratio |
|---|---|---|---|---|---|---|
| H5 | Y = SO + sex + age + rank + DC + rank*DC | 1,626.20 | 0.00 | 1.000 | 0.451 | 1.00 |
| H8 | Y = SO + sex + age + rank + DC + rank*DC + rank*sex + DC*sex + sex*rank*DC | 1,626.81 | 0.61 | 0.737 | 0.332 | 1.36 |
| H3 | Y = SO + sex + age + DC | 1,629.74 | 3.54 | 0.170 | 0.077 | 5.88 |
| H1 | Y = SO + sex + age | 1,630.10 | 3.90 | 0.143 | 0.064 | 7.02 |
| H4 | Y = SO + sex + age + rank + DC | 1,631.86 | 5.66 | 0.059 | 0.027 | 16.96 |
| H2 | Y = SO + sex + age + rank | 1,632.17 | 5.97 | 0.050 | 0.023 | 19.81 |
| H7a | Y = SO + sex + age + rank + DC + rank*sex | 1,633.32 | 7.12 | 0.028 | 0.013 | 35.16 |
| H7b | Y = SO + sex + age + rank + DC + DC*sex | 1,634.07 | 7.86 | 0.020 | 0.009 | 51.01 |
| H6 | Y = SO + sex + age + rank + DC + rank*age | 1,635.03 | 8.83 | 0.012 | 0.005 | 82.66 |
| H0 | Y = SO | 1,641.48 | 15.28 | 0.000 | 0.000 | 2,079.21 |

**Notes:**
Random effect: Cage; N = 234; SO, Sampling order; DC, Dominance certainty.

(w = 0.33, H8; Fig. 5). Specifically, for animals with more certain dominance relationships, an increase in rank of 0.25 was associated with 1.41 times *lower* levels of IL-6 (H5), an effect that may be stronger in males than in females (H8; effect size: 2.16 vs. 1.14, respectively; Table 3; Fig. 5). Among animals with less certain relationships, an increase in rank of 0.25 was associated with a 1.50 times (H5) *increase* in IL-6 with effects possibly being stronger in males than in females (H8; effect size = 2.11 vs. 1.24,

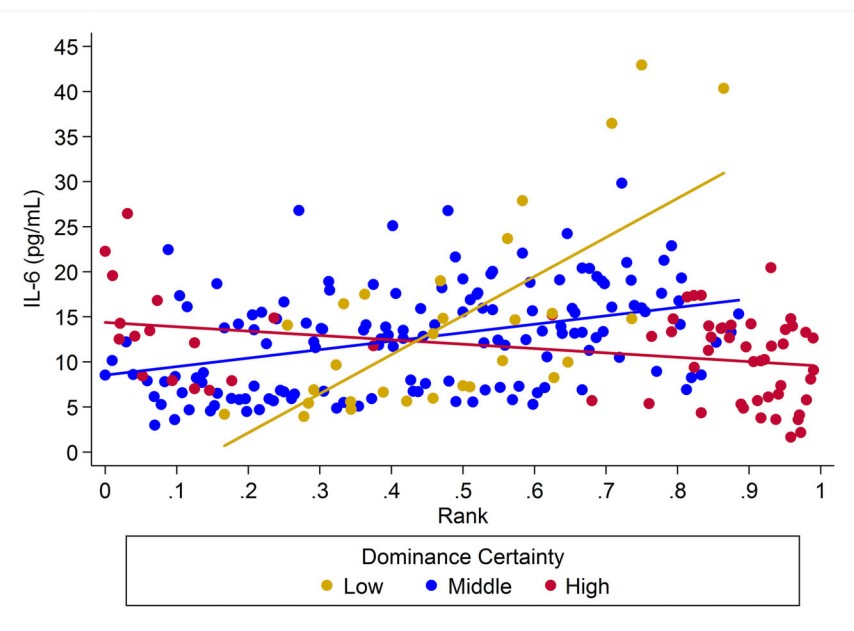

**Figure 4 The impact of dominance certainty and rank on inflammation based on serum levels of interleukin-6 (IL-6).** Predicted values of IL-6 based on model H5. Separate lines represent the interaction between dominance rank and dominance certainty.

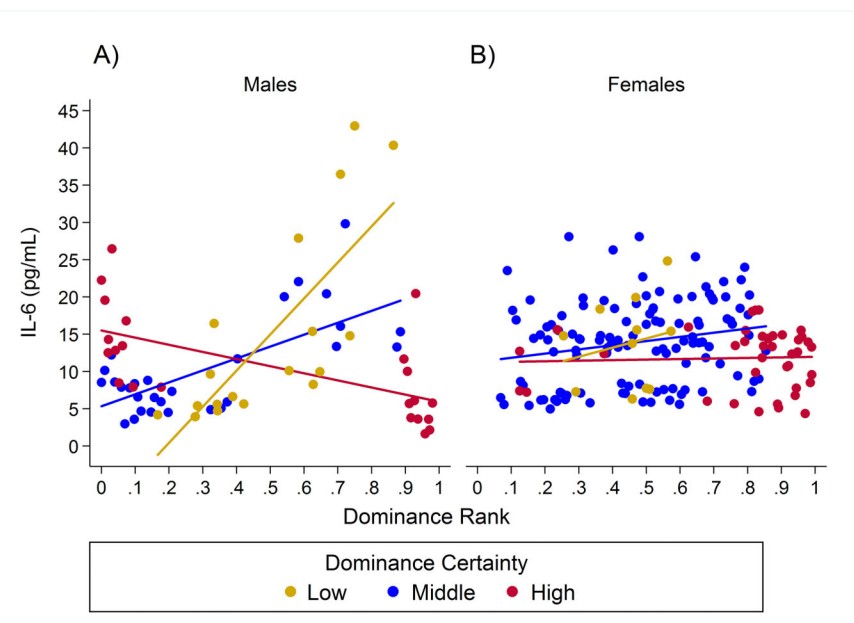

**Figure 5 Sex differences in the impact of dominance certainty and rank on inflammation based on serum levels of interleukin-6 (IL-6).** Predicted values of IL-6 based on model H8. Panels A and B depict effects for IL-6 for males and females, respectively. Separate lines represent the interaction between dominance rank and dominance certainty.

**Table 5 Model fitting for TNF-α.**

| | Model | AICc | ΔAICc | Model likelihood | Model weight | Evidence ratio |
|---|---|---|---|---|---|---|
| H5 | Y = SO + sex + age + rank + DC + rank*DC | 2,722.67 | 0.00 | 1.000 | 0.907 | 1 |
| H8 | Y = SO + sex + age + rank + DC + rank*DC + rank*sex + DC*sex + sex*rank*DC | 2,728.28 | 5.61 | 0.061 | 0.055 | 16.52 |
| H1 | Y = SO + sex + age | 2,731.06 | 8.39 | 0.015 | 0.014 | 66.39 |
| H0 | Y = SO | 2,732.14 | 9.47 | 0.009 | 0.008 | 113.64 |
| H3 | Y = SO + sex + age + DC | 2,732.42 | 9.76 | 0.008 | 0.007 | 131.30 |
| H2 | Y = SO + sex + age + rank | 2,733.22 | 10.56 | 0.005 | 0.005 | 195.88 |
| H4 | Y = SO + sex + age + rank + DC | 2,734.61 | 11.94 | 0.003 | 0.002 | 391.11 |
| H7b | Y = SO + sex + age + rank + DC + DC*sex | 2,735.81 | 13.14 | 0.001 | 0.001 | 713.37 |
| H7a | Y = SO + sex + age + rank + DC + rank*sex | 2,736.43 | 13.76 | 0.001 | 0.001 | 972.63 |
| H6 | Y = SO + sex + age + rank + DC + rank*age | 2,739.28 | 16.61 | 0.000 | 0.000 | 4,035.96 |

**Notes:**
Random effect: Cage; N = 234; SO, Sampling order; DC, Dominance certainty.

respectively). Taken together these results also indicate that high-rank animals with more ambiguous dominance relationships (i.e. low or moderate dominance certainty) exhibited higher levels of IL-6 than high-rank animals with more certain dominance relationships and that this effect may be specific to males.

Two simpler nested models were also part of the candidate set for IL-6 (Table 4). Model H3 (w = 0.08) included a main effect for dominance certainty plus age and sex terms, and indicated that an increase in dominance certainty of 0.10 was associated with 1.28 times lower levels of IL-6 (Table 3). Model H1 (w = 0.06) included only age and sex terms. For all four candidate models young animals (juveniles and subadults) had lower levels of IL-6 than adults (Table 3).

For the pro-inflammatory cytokine TNF-α, the model for H5 had the highest weight (w = 0.91; Table 5) indicating that the effect of rank on cytokine levels was dependent on dominance certainty for both males and females (Table 3; Fig. 6). For animals with more ambiguous dominance relationships (i.e. low dominance certainty) increasing in rank by 0.25 was associated with 2.07 times *higher* levels of TNF-α whereas increasing rank by 0.25 was associated with 1.78 times *decrease* in TNF-α for animals with highly certain dominance relationships (Fig. 6). Among low-ranked individuals, those with more certain dominance relationships had slightly higher levels of TNF-α compared to those with lower dominance certainty. Finally, juveniles and subadults had lower levels of TNF-α than adults (Table 3).

Plots of the raw data for CRP, IL-6, and TNF-α relative to rank and dominance certainty are presented in Figs. S1–S3.

## Diarrhea bouts

Results for diarrhea bouts showed no single model with highest weight (Table 6). The set of candidate models included H4, H3, H7b, H2, H5 and H7a. Dominance certainty was included as a main effect only (i.e. not part of interaction terms) in models H4, H3 and H7a and was predictive of the incidence of diarrhea in each of these models (Table 7).

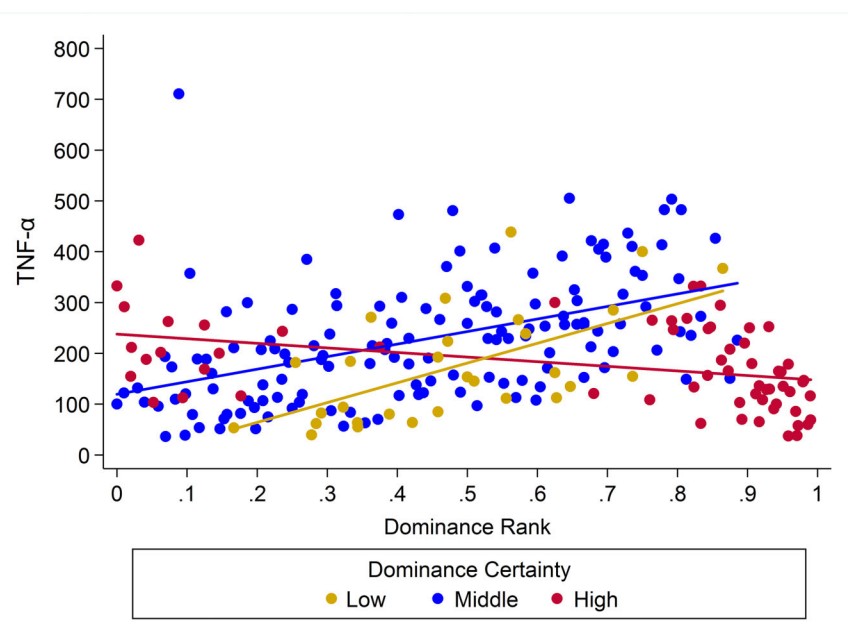

**Figure 6 Dominance certainty moderates the effect of rank on TNF-α.** Predicted values for TNF-α based on model H5. Separate lines represent the interaction between rank and dominance certainty.

**Table 6 Model fitting for diarrhea.**

| | Model | AICc | ΔAICc | Model likelihood | Model weight | Evidence ratio |
|---|---|---|---|---|---|---|
| H4 | Y = sex + age + rank + DC | 394.55 | 0 | 1.00 | 0.257 | 1.00 |
| H3 | Y = sex + age + DC | 394.65 | 0.10 | 0.95 | 0.245 | 1.05 |
| H7b | Y = sex + age + rank + DC + DC*sex | 396.17 | 1.62 | 0.45 | 0.114 | 2.25 |
| H2 | Y = sex + age + rank | 396.49 | 1.94 | 0.38 | 0.097 | 2.64 |
| H5 | Y = sex + age + rank + DC + rank*DC | 396.60 | 2.04 | 0.36 | 0.092 | 2.78 |
| H7a | Y = sex + age + rank + DC + rank*sex | 396.64 | 2.09 | 0.35 | 0.090 | 2.84 |
| H0 | Y = | 397.25 | 2.70 | 0.26 | 0.067 | 3.86 |
| H6 | Y = sex + age + rank + DC + rank*age | 400.08 | 5.52 | 0.06 | 0.016 | 15.81 |
| H1 | Y = sex + age | 400.09 | 5.54 | 0.06 | 0.016 | 15.92 |
| H8 | Y = sex + age + rank + DC + rank*DC + rank*sex + DC*sex + sex*rank*DC | 402.43 | 7.88 | 0.02 | 0.005 | 51.31 |

Notes:
Random effect: Cage; N = 252; Offset variable: days in cage; DC, Dominance certainty.

According to these models, reducing the certainty of one's dominance relationships by 0.10 (e.g., from 0.85 to 0.75) was associated with between 1.95 and 2.44 times *greater* incidence of diarrhea (Table 7), regardless of rank or sex. Models H5 and H7b were more complex, incorporating dominance certainty into interaction terms to determine whether its impact on health differed by rank or sex. The presence of these models in the candidate set suggest that the effect of dominance certainty might be dependent on rank or the effect of dominance certainty might be more pronounced in males (Table 7). However, addition
Table 7 Model coefficients from the set of candidate models for diarrhea bouts.

| | (H4) coeff (SE) | (H3) coeff (SE) | (H7b) coeff (SE) | (H2) coeff (SE) | (H5) coeff (SE) | (H7a) coeff (SE) |
|---|---|---|---|---|---|---|
| dAICc | 0 | 0.10 | 1.62 | 1.94 | 2.04 | 2.09 |
| Intercept | 0.16 (2.78) | 1.16 (2.73) | −1.30 (3.39) | −5.22 (0.50) | 2.19 (6.39) | −0.01 (2.84) |
| Rank | −0.96 (0.64) | – | −1.14 (0.69)[+] | 1.42 (0.60) | −4.28 (9.37)[*] | −1.09 (0.80)[+] |
| DC[1] | −7.00 (3.54)[*] | −8.90 (3.36)[*] | −5.10 (4.34) | – | −9.46 (7.81) | −6.71 (3.68)[+] |
| Sex[2] | −0.31 (0.35) | −0.27 (0.35) | 3.60 (5.31) | −0.27 (0.35) | −0.29 (0.35) | −0.47 (0.66) |
| Rank × DC | – | – | – | – | 3.96 (11.2) | – |
| Rank × Sex[2] | – | – | – | – | – | 0.36 (1.25) |
| DC × Sex[2] | – | – | −4.85 (6.60) | – | – | – |
| Rank × DC × Sex[2] | – | – | – | – | – | – |
| Age[3]-3 years | −1.09 (0.54)[*] | −0.90 (0.53)[+] | −1.09 (0.55)[*] | −1.13 (0.55)[*] | −1.11 (0.55)[*] | −1.09 (0.55)[*] |
| Age-4–5 years | 0.067 (0.36) | 0.20 (0.35) | 0.09 (0.36) | −0.05 (0.36) | 0.07 (0.36) | 0.08 (0.36) |
| Age-13+ years | 0.65 (0.42) | 0.62 (0.42) | 0.69 (0.43) | 0.60 (0.43) | 0.63 (0.43) | 0.64 (0.42) |

Notes:
[1] DC, Dominance certainty.
[2] Males were referent category.
[3] Adults (6–12 years) were the referent category.
[*] $p < 0.05$.
[+] $p < 0.1$.

of these interaction terms did not improve model fit, as AIC increased by 1.6 and 2.0 in H7b and H5, respectively (Table 6).

Dominance rank was included as a main effect only in models H2, H4, and H7b (Table 6). According to these models, reducing dominance rank by 0.25 (e.g. from outranking 75% to outranking 50%) was associated with between 1.25 and 1.42 times *greater* incidence of diarrhea (Table 7). Models H5 and H7a incorporated dominance rank into interaction terms with dominance certainty (described above) and with sex. Addition of the dominance rank × sex interaction did not improve model fit and AIC increased by 2.09 (Table 6). As expected, older animals showed a higher incidence of diarrhea bouts than adults and juveniles across all models (Table 7).

## DISCUSSION

A wealth of literature highlights the fact that uncertainty is a potent stressor and can lead to a wide range of negative health outcomes (*Baum & Fleming, 1993*; *Heaney, Israel & House, 1994*) highlighting the potential importance of variation of dominance certainty in studies of social status. The results we present demonstrate that more complex conceptualizations of social status provide a possible explanation for seemingly contradictory findings in the status-health literature across species. The health risk for any given animal is dependent on a combination of both the individual's absolute status and a metric of the certainty or stability of that status.

The measure of relationship uncertainty used here, an average of all dyadic level dominance certainty, can also potentially offer a metric of how well an individual "fits" within the dominance hierarchy overall. Low average dominance certainty could reflect an animal that is changing in position within the hierarchy (moving up or down), or, due to

the use of direct and indirect pathways in a network, low dominance certainty could also arise for an individual because others in their local community are changing their position(s) in the hierarchy. For example in Fig. 1B, we can see that there is ambiguity in the likely rank relationship between animals F and I, not because they are in a direct contest with each other, but due to inconsistency in the direction of the flow of dominance in their local community (i.e. information inferred from the blue and red arrows is contradictory). A network approach can examine how instability in dyadic relationships might ripple throughout a local community and potentially impact the position of nearby neighbors in the network.

## Rank and dominance certainty

Variation in social status, and thus its potential impact on health, includes not only whether one has high or low status but also the relative predictability or certainty of one's status relationships. Dominance rank and dominance certainty showed a nonlinear (U-shaped) relationship where individuals of both high and low dominance rank had relatively greater dominance certainty than individuals in the middle of the hierarchy. However, variability in dominance certainty at all levels of dominance rank suggests that these two metrics may describe two complementary aspects of social status that can have independent and interacting effects on an animal's health. The idea that dominance rank stability can have a critical impact on the status-health relationship has been suggested previously (Sapolsky, 1992; Sapolsky, 2005; Marmot & Sapolsky, 2014). The current paper, however, expands on previous research by using a more general measurement of each individual's "fit" within the group hierarchy, as opposed to examining the status-health relationship in stable vs. unstable groups, or only in individuals that experienced dominance rank reversals.

## Status and measures of inflammation

Our research suggests that whether individuals have high or low dominance rank in the hierarchy does not fully represent the complexity with which social status affects health. Instead, dominance certainty modified the impact of rank on biomarkers of inflammation. Under conditions of low dominance certainty, we found that high-ranking animals had higher levels of pro-inflammatory cytokines and CRP. This is consistent with previous findings in African wild dogs, baboons, chimpanzees, dwarf mongooses, and ringtail lemurs (Creel, MarushaCreel & Monfort, 1996; Cavigelli, 1999; Muller & Wrangham, 2004; Gesquiere et al., 2011), where high dominance rank individuals (or at least the alpha individual, as in Gesquiere et al. (2011)) had elevated GCs. Because dominance certainty is quantified as the consistency in the flow of dominance between pairs in the aggression network, having low certainty may be evidence of an individual-level tendency to protest or challenge others' dominance rank or precursors of a rank change (within a hierarchy that is stable overall). Thus, high-ranking animals with ambiguous dominance relationships likely have reduced predictability that others will submit to them and greater potential risk of losing status (e.g., Crockford et al., 2008), and this type of uncertainty could represent a major psychosocial stressor. Our findings

are consistent with previous research indicating that instability, whether due to dominance style (e.g. chimpanzees, *Muller & Wrangham, 2004*) or due to current social factors (e.g. baboon dominance rank reversals, *Sapolsky, 1992*), is particularly bad for high-ranking individuals, possibly due to potential for loss of status. For example, in an examination of rank reversal in male baboons, *Sapolsky (1992)* reported that despite similar levels of participation in aggressive interactions, only males about to lose their rank showed elevated GCs, not those about to rise in rank.

In contrast, for animals with a high degree of certainty in their dominance relationships, we found a small reversal of the effect with high-ranking animals having slightly lower levels of inflammatory markers than low-ranking animals. This effect is weak, but notably, more closely matches results from humans, baboons, macaques, meerkats, and spotted hyenas in which low status individuals exhibited risky health profiles (*Sapolsky & Share, 1994*; *Shively & Clarkson, 1994*; *Goymann et al., 2001*; *Young et al., 2006*; *Ostner, Heistermann & Schülke, 2008*; *Gesquiere et al., 2011*; *Marmot & Sapolsky, 2014*). In contrast to high status individuals, where uncertainty may have been associated with a potential loss in status, dominance uncertainty may not be a stressor for low-status individuals because they have little to lose; it may even be the case that low-ranking individuals with low dominance certainty may have an opportunity to increase their status—a possibility that needs to be explored further in future studies. Among low-ranking animals, being certain of one's low dominance rank is likely to be stressful because such individuals can reliably expect to receive aggression, harassment, and/or intimidation from dominants, have little control over the occurrence of such interactions, and have fewer social outlets to cope with this harassment (*Schino, 2001*; *Sapolsky, 2005*). This is largely consistent with Shively & Clarkson's findings that female macaques that were experimentally arranged to lose status (via group membership manipulations) increased atherosclerosis by 500%, whereas those that gained status increased atherosclerosis by a far smaller amount—44% (*Shively & Clarkson, 1994*). Thus, dominance rank changes may be stressful for both parties, but losing status is likely worse than gaining status.

## Sex differences

Although sex differences in the status-health relationship are not often studied, sex differences have been noted previously (*Kaplan & Manuck, 1999*). Our results for the pro-inflammatory proteins, CRP and IL-6, suggest that there may be sex differences in the impact of rank on inflammation. Elevated levels of CRP in high-ranking individuals with lower dominance certainty was found only in males, not in females. The lack of an effect for females of this species is not surprising given the behavioral biology of rhesus macaques, specifically due to sex differences in how dominance rank is gained and maintained. Rhesus macaque males emigrate, and once they are established in a new group, they can increase their rank through both alliances and direct competition. Female macaques, on the other hand, remain in their natal group and inherit their dominance rank from their mothers with all the females of one family outranking all the females from another family (*Lindburg, 1971*; *Sade, 1972*; *Melnick, Pearl & Richard, 1984*). This process results in a male hierarchy that is more labile and changeable, but changes in

dominance rank are rarer for females (*Sade, 1972*; *Berard, 1990*). For females, due to inherited rank and lifetime tenure in social groups, we would expect to see little dominance uncertainty, except during periods of larger scale group instability, such as when one matriline threatens to overthrow another matriline (*Ehardt & Bernstein, 1986*; *Beisner et al., 2011*).

### Diarrhea

Dominance uncertainty was a better predictor of diarrhea than dominance rank with lower certainty being associated with greater risk of diarrhea. While animals with lower dominance certainty may be more vulnerable to diseases causing diarrhea, stress is also a known contributor to diarrhea. It may be the case that the increased stress of uncertainty in social relationships is contributing to the incidence of non-pathogenic diarrhea (e.g., *Stasi et al., 2012*; *Buckley, O'Mahony & O'Malley, 2014*). The small effect of dominance rank for diarrhea is consistent with other studies in baboons, macaques, meerkats, and spotted hyenas that demonstrate a wide range of negative biomarkers of health, as well as poor health outcomes in low-ranking individuals (*Sapolsky & Share, 1994*; *Shively & Clarkson, 1994*; *Goymann et al., 2001*; *Young et al., 2006*). Notably, unlike the biomarkers of inflammation discussed above, there was not strong evidence for an interaction between dominance rank and dominance certainty. This result suggests that the impact of social status on health is specific to the type of status measure being examined (e.g. certainty vs. rank) as well as the particular health or fitness outcome of interest (e.g. biomarkers of inflammation).

## CONCLUSION

Our research demonstrates the importance of more complex representations of social status for understanding its impact on health. Our data show that the effect of social status on health is much better understood by accounting for status certainty. Indeed, the interaction between a linear measure of status and status certainty in our data reconciles the contradictory patterns in dominance rank and health found in the previous literature as a direct result of framing social status in terms of its certainty. Our work suggests that expanding the examination of the certainty of social relationships, or fit within one's social class, may be a critical step toward understanding status effects on health outcomes. As such, the innovative methods leading to this more complex conceptualization of status, as presented here, promises to significantly enhance our ability to detect more effectively who may experience health related costs in society.

Our results also demonstrate that computational social network techniques have the capacity to advance our understanding of the impact of social status on health by disentangling the relative effects of linear measures of rank versus individual-level uncertainty of rank relationships. To the best of our knowledge, we are the first to empirically demonstrate that uncertainty in an individual's dominance relationships, as measured by inconsistency in the direction of one's dominance network pathways, are associated with multiple indicators of poorer physical well-being. In contrast, the impact of dominance rank on pro-inflammatory proteins was dependent upon dominance

certainty, suggesting that the effects of social status on health are highly dependent on the context in which they occur.

## ACKNOWLEDGEMENTS

We would like to thank our dedicated team that collected the behavioral and physiological data (A. Barnard, T. Boussina, A. Vitale, E. Cano, J. Greco, N. Sharpe, S. Seil, J. Carabez, R. Pisano, I. Wong, K. Balasubramaniam, H. Schwertscharf, C. Bonilla). We would also like to thank the reviewers of this manuscript for their valuable comments and feedback.

### Funding

This research was funded by an NIH grant awarded to Brenda McCowan (R01-HD068335) and the California National Primate Research Center base grant (P51-OD01107-53). The funders had no role in study design, data collection and analysis, decision to publish, or preparation of the manuscript.

### Grant Disclosures

The following grant information was disclosed by the authors:
NIH grant awarded to Brenda McCowan: R01-HD068335.
California National Primate Research Center base grant: P51-OD01107-53.

### Competing Interests

The authors declare that they have no competing interests.

### Author Contributions

- Jessica J. Vandeleest conceived and designed the experiments, performed the experiments, analyzed the data, wrote the paper, prepared figures and/or tables, reviewed drafts of the paper.
- Brianne A. Beisner conceived and designed the experiments, performed the experiments, analyzed the data, wrote the paper, prepared figures and/or tables, reviewed drafts of the paper.
- Darcy L. Hannibal conceived and designed the experiments, performed the experiments, analyzed the data, wrote the paper, prepared figures and/or tables, reviewed drafts of the paper.
- Amy C. Nathman performed the experiments, wrote the paper, reviewed drafts of the paper.
- John P. Capitanio conceived and designed the experiments, contributed reagents/materials/analysis tools, wrote the paper, reviewed drafts of the paper.
- Fushing Hsieh conceived and designed the experiments, contributed reagents/materials/analysis tools, reviewed drafts of the paper.
- Edward R. Atwill conceived and designed the experiments, contributed reagents/materials/analysis tools, reviewed drafts of the paper.

- Brenda McCowan conceived and designed the experiments, contributed reagents/materials/analysis tools, wrote the paper, reviewed drafts of the paper.

## Animal Ethics

The following information was supplied relating to ethical approvals (i.e., approving body and any reference numbers):

This research was approved by the University of California Davis Institutional Animal Care and Use Committee.

## Data Deposition

Dash (https://dash.cdlib.org/) DOI 10.15146/R3V883; http://n2t.net/ark:/c5146/r3v883.

## Supplemental Information

Supplemental information for this article can be found online at http://dx.doi.org/10.7717/peerj.2394#supplemental-information.

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
