# Peer review of "Decoupling social status and status certainty effects on health in macaques: a network approach"

_PeerJ, doi:10.7717/peerj.2394_

## Round 0.1 · original submission · Major Revisions

· Academic Editor

Major Revisions

Dear Dr Vandeleest,

Thank you for your submission, which has now been seen by two reviewers, both of whom offer some detailed and constructive comments. Based on their assessment and my own reading of your paper, I would like you make some major revisions of your manuscript. These mainly concern providing more detail of your network based approach, and there are certain aspects of your statistical approach that may need some rethinking and refining. In particular I agree with reviewer 1 that, if using an information theoretic approach, you should present all models, and your interpretation should not be based on p-values, nor talk about significance. The rationale for choosing particular kinds of models could also be better explained as reviewer 2 suggests.

I hope you will find these comments useful, and I look forward to seeing a new version of your paper soon.

With best wishes,

Louise

·

Basic reporting

This manuscript investigates the effect of dominance rank and uncertainty on several health outcomes in three big captive groups of rhesus macaques by using an analytical tool derived from social network analysis. The study shows that dominance rank alone does not account for blood levels of inflammatory response and the symptom of diarrhoea but its effect is mediated by dominance uncertainty, i.e. how consistent individual’s agonistic behaviour (given and received) is. A novel computational network-based approach called “percolation and conductance” is used to assess dominance uncertainty. The topic investigated is timely and interesting, the study aims are relatively straightforward and the methods have a high level of technicality yet seem to be practical.

Experimental design

My main issues with this manuscript are two-fold. One issue is mainly technical. First, although using a network approach helps taking into account so called indirect relationships or influences, there is no real effort at explaining how this novel approach compares to others for example looking at the transitivity of relationships. The applicability and functionality of this method compared to others could be fleshed out a bit more. Then, network measures are typically not independent, specifically and most particularly indirect measures because a change in one node even several connections away can still change the focal node’s measure. If the percolation and conductance method accounts for this, it would be important to mention it. If not, how would this issue be addressed? (or if dismissed, with which arguments).
The other issue is more problematic and is statistical. I disagree with the use of an information-theory approach together with model selection and p-values. Using AIC-based inference is laudable but should be done properly, i.e. unconditionally. This means that all models should be presented together with all dAIC (or DIC), weights, evidence ratio and relative likelihoods and the results should present that without discussing the “statistical significance” of a variable. For example, the results can be discussed in terms of how much of the variance is explained or of the likelihood of the relationship going in one direction compared to the other (see the special issue of Behavioural Ecology and Sociobiology in 2011 "Model selection, multimodel inference and information-theoretic approaches in behavioural ecology" Volume 65, Issue 1, including Burnham, Anderson, and Huyvaert, “AIC model selection and multimodel inference in behavioral ecology: some background, observations, and comparisons” 65:1, 23–35, and Mundry “Issues in information theory-based statistical inference-a commentary from a frequentist’s perspective” 65: 1, 57–68; See also Mundry and Nunn 2009 “Stepwise model fitting and statistical inference: turning noise into signal pollution.,” Am. Nat. 173: 1, 119–123 and examples of work citing these references). Related to that, lines 228-230, it is written “Effects of age for the IL-6, TNF-α, and CRP models did not mask or alter the effects of other variables presented in any models and were therefore excluded from the analyses to preserve power.” It contradicts the use of an IT approach and more importantly, if age has been thought as a factor important enough to be included in the first place, even if it is not contributing “significantly” to the models, it must stay in the models as it is part of the structure of the data and of the models. Otherwise, this could look like stepwise model selection, which is also quite controversial (see above articles).

Validity of the findings

The present manuscript certainly has an added value compared to previous studies on the same topic and can inspire other areas of research.

Additional comments

Finally, the manuscript needs to be edited a bit so as to either avoid typos and repetitions or be more explicit. Here are a few examples:
Lines 61-64: I am not convinced that these sentences belong here because some concepts such as social status haven’t been defined yet. Maybe that could be phrased more generally?
Line 71: indicate in what way GC levels are indicators of health outcome.
Line 85: it would be useful to precise that it concerns only males.
Line 123: I would use “long-tailed macaques”
Line 124: add “than high ranked individuals”
Line 127: as the link between health outcome and stress hasn’t been made in the introduction, I would suggest either to add a few more words here to complete the argument or to stop the sentence at outcomes.
Line 128: outcome singular. it is gross enough...
Lines 175-177: could you provide a rational for doing so?
Lines 180-181: was it equivalent for all groups? In general is there any group difference at all?
Lines 202-214: please define shortly what IL-6 and TNF- and C-reactive protein assay are and how they relate to poor or good health outcomes and give references to previous works to help readers properly understand how this part of health outcomes was assessed.
lines 243-244: could you be a bit more specific?
line 303: you mean chimpanzees? I would also put the species and the reference together instead of separately.
Line 345: outcome singular
Lines 346-357: I am not sure this paragraph fits under the headline “gross health outcome”. It would certainly benefit to have its own paragraph in a more general discussion, while the rest of it (lines 358-369) can be added before line 346.
Figure 4: could it be in the same format than Figure 3?

·

Basic reporting

The authors present a unique novel approach to examining the link between social status (i.e., dominance rank) and health outcomes (inflammation and diarrhea) using captive rhesus macaques as a model system. Interestingly, they found that dominance certainty (i.e., “the relative certainty vs. ambiguity of an individual’s status”) moderates the effect of social status on health outcomes. Overall, the paper is well written, theoretically grounded, and an extremely valuable contribution to the literature. However, I have three major concerns with the current manuscript. First, the methods, in particular the models and their novel approach to calculating dominance certainty, should to be described in much more detail. Second, the authors combine adults and juveniles of both sexes into one dominance hierarchy, which may not accurately reflect how rank affects individuals of different ages and gender. Third, plotting the predicted values from your models as data points can be deceptive. The authors should address these and other concerns (detailed below) prior to acceptance.

Experimental design

Description of methods:
A crucial component of this manuscript is the novel network-based approach to calculating both dominance rank and “dominance certainty” (DC). However the description in lines 157-170 is hard to follow. If this method is described in another paper (or, in fact, in the R package of the Fujii et al, 2015 reference), then the authors do not need to spend so much time explaining how DC is calculated. In fact, in its current form, the description is somewhere in the middle – either too descriptive or not descriptive enough. For example, if you are going to casually mention “percolation algorithms”, “flow pathways”, and the “conductance principle” then it would be good to clearly explain them. Alternatively, if these details are not essential to the paper, then the authors could just say “we calculated dyadic dominance certainty using the R package “perc” and averaged this for each individual. These numbers range from 0.5 to 1, with…”
The confusion of this description is compounded by the fact that the authors use a lot of jargon, such as alternating between “individual” with “node”. As is, this part of the paper is highly technical and difficult for an animal behavior (or health) researcher to understand. A very specific example of a confusing (and circular) sentence is that in Lines 173-175, which currently reads as “an animal’s dominance certainty was used as the certainty of their dominance rank”.
Why certain generalized linear mixed-effects models were chosen could be described in a more detail. Rather than just stating that the authors looked at the distributions to determine which model family to use (lines 220-221), it would be nice if these plots were shown in the supplemental material. I downloaded the data and plotted the distributions for CRP, TNF, and IL6 and, to me, it looks like they could all (potentially) be modeled using the same family. The author’s decision to model two IL6 and TNFα as a negative binomial and CRP as a gamma seems rather arbitrary. Also, if this is how the authors are fitting their models, then why are the data presented with what looks like a linear fit in all of the figures? If possible, the authors should do some sort of transformation of the data (a log transform might work) and then model those transformed outcome variables using a linear mixed-effect model. Would the results still remain the same? The choice to use a Poisson for modeling diarrhea is also puzzling since the majority of individuals have a score of zero here. Perhaps the authors could think about using a zero-inflated Poisson?
Finally, in lines 228-230, the authors report that the effect of age for the three biomarker models did not “mask or alter the effects of other variables and were therefore excluded”. Could the authors please show these results? In my brief look at the data, I found a correlation between age and rank (r=-0.33), with higher-ranking animals being older, as well as a smaller correlation between age and DC (r=0.18). Further, the correlation between rank and age is much stronger in males than it is in females (rank v age correlation in males = -0.68). Given these correlations and the fact that we know that age influences inflammation, it would be best if the authors kept age in as a covariate in all of their models. This would only reduce the degrees of freedom by 1, since age can be modeled as one continuous variable (another question I had was why the authors used 4 age categories instead of a continuous measure of age, which would use up 2 fewer degrees of freedom).

Estimating dominance rank for all ages and genders together:
As the authors mention, there are “sex differences in how rank is gained and maintained” (line 335). This is also the case for age: rank means different things for infants and juveniles when compared to adults. I feel like the results could change if the authors included separate dominance ranks for adult females (relative to other adult females) and adult males (relative to other adult males). Could the authors demonstrate the robustness of their findings by only looking at adult males (or adult females)? Right now it is hard to determine if the effect of rank and DC are independent of their relationships with age.

Plotting predicted values:
The captions state that the “predicted values” of each variable are plotted. This can be very deceiving because it assumes that your models are correctly capturing the data (which is not necessarily the case). It would be much more accurate to plot the observed values and then show the best fit line through those data points. If that looks too messy, then the authors could also plot their predictors of interest (DC or rank) against the residuals from a reduced model that doesn’t include these predictors.

Validity of the findings

see comments on the methods (above)

Additional comments

I have a few more minor comments:
Lines 293-294 (and elsewhere): The authors should reference Gesquiere et al, 2011 (doi: 10.1126/science.1207120), when mentioning how high ranking males have higher GCs. Gesquiere’s showed that the alpha male had the highest GC levels, which were comparable to the lowest-ranking males.

Line 180: Did the authors mean to write “the segmented package in R” rather than “packed”? Also, please italicize any R packages in the text.

Line 284-286: This is a really great statement and is probably the reason why this work is such a great addition to the literature. Maybe this could be moved up to the very first paragraph in the discussion?

Lines 348-357: This is a very nice description of dominance certainty. I think the authors should take this description and move it up into the introduction (or even the methods!). It feels a bit out of place here at the end of the discussion and it is relevant to the whole paper.

The 95% CI lines in the figures make them a bit busy. These might be unnecessary and can be removed since the values are being plotted (now the observed values),

---

## Round 0.2 · Minor Revisions

· Academic Editor

Minor Revisions

Thank you for your revision, which I sent back to our previous reviewers, and I have now read over the MS carefully myself. As both reviewers note, you have done an excellent job addressing their concerns, and your paper is almost ready for publication. As you will see, however, both reviewers have a few remaining concerns, mostly associated with the analyses and presentation of your results, and I agree that making these changes would further strengthen the paper. In particular, I think reviewer 1's suggestion for model averaging makes a lot of sense, in terms of being able to interpret your findings and get a good grasp of what's going on here. I have therefore recommended minor revisions, and would like you to consider these further suggestions. I think they are intended fully constructively, and I hope you will take them in this spirit, even though they require a little more work from you and your co-authors.

with best wishes,
Louise

·

Basic reporting

No comments

Experimental design

Concerning the description and overall usage of the I-T approach: although for some of the dependent variables, one model clearly stands out as one with more explanatory values compared to all others (dAIC > 10), I would still use model averaging to obtain data on single variables because interpreting the results should still incorporate the uncertainty in the data. Using model averaging, whether full or partial, providing average parameters estimates, unconditional standard errors and 95% confidence intervals as well as normalized Akaike weight for each variable is as useful, if not more, to assess a variable’s contribution and importance to explaining the dependent variable as providing p-values. Providing only p-values in the text while giving the results is not informative at all so I would remove them and either refer to the table or give all characteristics between parentheses (estimate +/- SE, weight, CI). I have nothing against p-value per se, but in that particular case, I think it could be misleading and prone to too much estimation error. In addition, averaging allows to have one “result” for each variable regardless of how many models contained this variable AND are within the most parsimonious model pool (so what is explained lines 287-289 would not be necessary for example). I like the fact that effect sizes are mentioned and discussed. Finally, when presenting the results in Table 2, 4, 5 and 6, for the sake of clarity, it would be better to order the models according to the difference in AICs with the most parsimonious model first. It would also be useful to know the cutoff point regarding dAIC: is it 2, 4, 7, 10 (see Burnham, Anderson & Huyvaert 2011 BES)? And I would also provide evidence ratio (weight of the most parsimonious model divided by the weight of the model to compare) as this gives information about how much more support this “best” model has compared to others. If the authors need guidance as to how best to present their results, a good recent example is Wagner et al. "Temporal comparison and predictors of fish species abundance and richness on undisturbed coral reef patches." PeerJ 3 (2015): e1459, 10.7717/peerj.1459.

Concerning the use of a zero-inflated model for diarrhoea, an issue that is raised by the second reviewer, I tend to agree with the authors that their data doesn’t necessarily warrant modelling zero-inflation. However, given the distribution, nature and assumption of the data, it could be modelled following a negative binomial.

Others:

Concerning the part “Social Network Measures and Independence”: please give details about the number of times you performed the removal-recalculation procedure and give that number with the correlation results.
Line 214: could you precise an average of what?
Line 303: what is the unit of measurement of IL-6 and TNF-alpha? As far as I can see, they are not discrete data type so how can this be modelled with a discrete distribution?
Line 379: no numerical result or reference to a table?
Put Table 3 as Table 6 and Table 4, 5, and 6, as Table 3, 4 and 5.

Validity of the findings

No comments.

Additional comments

The authors did a great job answering the issues that were raised and substantially improved their ms.

·

Basic reporting

The authors have addressed many of my concerns in this thoughtful revision and I still believe that the paper is well written, theoretically grounded, and an extremely valuable contribution to the literature. The authors did a wonderful job of describing their method for calculating indirect “wins” and dominance certainty. I also appreciate the more detailed comparison of hypotheses and the author’s comprehensive IT-based approach. I only have one large comment on their choice to plot predicted value and a few smaller comments that the authors should address prior to acceptance:

Experimental design

No comments

Validity of the findings

Plotting predicted values:
I still feel that the authors should not be plotting the predicted values from their GLM. These numbers can be deceiving because it assumes that your models are capturing the data with little error. If you think of the simplest, bivariate model where you modeled y as a function of x. If you decided to plot the predicted values of y then they would fall completely on the best linear fit of x on y. But your true data points would not fall on top of that line (unless the model was a perfect fit). Thus, I still recommend that the authors plot the observed values, even if it does not look as pretty. It is a more appropriate way of presenting the data. As I also mentioned in my first review: the authors could plot their predictors of interest (dominance certainty or rank) against the residuals from a reduced model that doesn’t include these predictors. In their response the authors mentioned that “Because multiple factors contribute… [this approach] would fail to capture important variation in the data”. I disagree with this sentiment. The idea behind plotting the residuals from a reduced model (that does not include dominance certainty or rank) is that it would control for the other factors that contribute to the variance in your outcome variables. If the authors do not approve of this approach, then I would suggest removing Figures 3 and 4 or coming up with an alternative that does not include predicted value.

Additional comments

Minor comments:
L66-67: An appropriate, hot-off-the-press reference that should be added here is Chetty et al, JAMA, 2016 (doi: 10.1001/jama.2016.4226)

L203: Did you mean “only 42.3% of all possible dyads had at least one agonistic interaction in the network.” ?

L205: I don’t think that you want to use the word “fight” here, since many of your dominance behaviors are non-contact threats (or submissions).

L210-212: I would either remove this sentence or make it fit more organically within the previous paragraph. Right now it reads as an addendum.

L218-220: “We also transformed ordinal dominance ranks for each group into the proportion of animals outranked within their respective groups”
Was this done using the matrix with certainty measures, which ranged from 0-1? Or did you calculate ordinal rank another way and the matrix only had 0s and 1s? One way to get the “proportion of animals outranked” that incorporates your uncertainty would be to take the average across the rows of your dominance certainty matrix (the one that you get from the Perc package with values that range from 0-1). Do those values match your ordinal ranks?

L224-226: how different were the ranges of “low” “medium” and “high” dominance certainty across the three groups?

L338-344: I would remove some of the modifiers (e.g., “potentially”, “may be”, and “possibly”) from your descriptions about the sex-rank interaction effects since you do not use those modifiers for the dominance certainty effects, which may not be as strong as the sex effects.

L419: thank you for adding the Gesquiere reference here, but I think it has been misinterpreted. They found that the alpha male had high GCs (comparable to the lowest ranking males), but that the 2nd ranking male had the lowest GCs. So it was the fact that the alpha male was likely to be in a tenuous position where he was constantly fighting off competitors. Also, it was in baboons, not wild dogs, chimps, mongooses, or lemurs.

Table 3: Please show the standard errors for the coefficients as well as the full p-values (not just “<0.05” and “<0.1”). It will help the reader to understand the strength of the effects of each predictor.

Figure 3: please label the x-axis “dominance rank”

---

## Round 0.3 · accepted · Accept

· Academic Editor

Accept

Thank you for your revision, and for your comprehensive response to the reviewers' comments. Having read up a little more on model averaging myself, based on what you say here, I agree that your approach is more appropriate (and learned a lot in the process!), and I think you've dealt with all the other comments appropriately as well. I am therefore delighted to accept your paper for publication.
Congratulations on a fine piece of work--I hope it generates a lot of interest!

all the best,
Lou